# Shared Visual Representations of Drawing for Communication: How do different biases affect human interpretability and intent?

**Daniela Mihai**
Electronics and Computer Science
The University of Southampton
Southampton, UK
adm1g15@soton.ac.uk

**Jonathon Hare**
Electronics and Computer Science
The University of Southampton
Southampton, UK
jsh2@soton.ac.uk

## Abstract

We present an investigation into how representational losses can affect the drawings produced by artificial agents playing a communication game. Building upon recent advances, we show that a combination of powerful pretrained encoder networks, with appropriate inductive biases, can lead to agents that draw recognisable sketches, whilst still communicating well. Further, we start to develop an approach to help automatically analyse the semantic content being conveyed by a sketch and demonstrate that current approaches to inducing perceptual biases lead to a notion of objectness being a key feature despite the agent training being self-supervised.

## 1 Introduction

In multi-agent emergent communication research, one of the long-term goals is to develop machines that can successfully communicate between themselves but also with humans. Visual communication in the form of sketching and drawing, which has long preceded written language communication, can directly be interpreted by a human observer (Gelb, 1963; Eitz et al., 2012). Recently Mihai & Hare (2021b) demonstrated that it was possible to train a pair of agents parameterised by artificial neural networks to play a visual communication game by communicating through line drawings. Such a demonstration was often discussed as a logical next step in the emergent communication community but was challenging until the innovation of a differentiable rasterisation algorithm (Mihai & Hare, 2021a) that allowed end to end training. The referential signalling games, inspired by Lewis (1969), involved a 'receiver' agent that was presented with a set of images, and a 'sender' agent that was presented with a single image from the receiver's set. The goal of the game was for the sender agent to communicate to the receiver which image they had by drawing a picture using a fixed number of line strokes. Without specific inductive biases, it was demonstrated that successful communication could be achieved between the agents, however, the images themselves were not interpretable; they were in essence the visual equivalent of a hash code. Mihai & Hare (2021b) however, went on to further demonstrate that, with appropriate representational biases, induced by additional loss functions during training, there was strong evidence that the drawings produced by the machine could also be used to communicate successfully with humans.

In this paper, we investigate the factors that lead to shared visual representations of drawing between humans and machines. More concretely, we explore the effect of different perceptual losses and visual encoders on making sketches produced by a drawing model (Mihai & Hare, 2021b) more interpretable to a human observer. We start by introducing a more powerful network for encoding visual information — the pretrained Vision Transformer (Dosovitskiy et al., 2021) from the CLIP framework (Radford et al., 2021), and then explore different approaches to inducing the network to

3rd Workshop on Shared Visual Representations in Human and Machine Intelligence (SVRHM 2021) of the Neural Information Processing Systems (NeurIPS) conference, Virtual.

produce more understandable drawings. We compare against the pretrained VGG16 feature extractor used in the original model and develop an approach that enables us to ask what the main semantic content of the drawings are using "prompt engineering" (Radford et al., 2021) with the CLIP model.

## 2 Background

Marr's classical theory of vision (Marr, 1982) posited that sketch-like representations were a fundamental part of human vision. Works from Neuroscience also suggest that simple line drawings capture the core visual features needed to recognise physical objects (Biederman & Ju, 1988) and activate the same brain regions used to distinguish natural scene categories from colour photographs (Walther et al., 2011). Recent works on visual communication games (Mihai & Hare, 2021b; Fernando et al., 2020), which are analogous to the approach taken by emergent communication research (Havrylov & Titov, 2017; Lazaridou et al., 2017, 2018; Bouchacourt & Baroni, 2018) have shown that agents can successfully convey information by drawing, which is a much simpler and more interpretable form than language. The main challenges with a language-based communication protocol developed between agents playing such games are its interpretability and grounding into natural language, which would allow a human to understand and use the information conveyed (Lowe et al., 2019). Drawing and sketching, on the other hand, have been used since prehistoric times by humans to depict the surrounding visual world, long before the emergence of written language (Gelb, 1963). Sketches of a visual scene produced by a human could have various levels of abstraction: from very detailed and close to reality representations, to depictions of just the semantics expressed in terms of the objects in the scene and their relations, to low-level representations like edges and key points. This work, however, looks at the factors that can make a *constrained* visual communication channel interpretable for human observers.

## 3 Extended Model with CLIP

In this work, we follow the game setup proposed by Mihai & Hare (2021b) which was inspired by Havrylov & Titov (2017)'s image guessing game. As illustrated in Figure 1, the game requires the sender to communicate the target image to the receiver, by sketching 20 black straight lines. The receiver has to guess the correct image from a pool of photographs consisting of $K$ distractors plus the target. The model is trained end-to-end with a multi-class hinge loss, we refer to this game objective as $l_{game}$. However, the addition of a perceptual loss, $l_{perceptual}$, has been shown to improve humans ability to recognise the object depicted in the sketch (Mihai & Hare, 2021b). An overview of the agents' architecture and game setup is shown in Figure 1.

In the "original" game setup, the photograph that the sender communicates about matches the target from the receiver's pool of images. Mihai & Hare (2021b), however, proposed other game setups in which this requirement does not hold. For the purpose of this study, we only explore the original game variant for which we set the number of distractors, $K$, to 99. It is worth noting that this sort of game would be very difficult for humans to play. Guessing the target from a set of 100 images, which could contain multiple examples from the same class as the target, based only on a 20-line black and white sketch seems impossible for humans. The trained agents, however, manage to establish a visual communication protocol that can be used to successfully solve the task as shown in Section 3.2.

### 3.1 Inducing interpretable drawings with "perceptual" losses

Zhang et al. (2018) demonstrated that a loss computed using the weighted difference of features extracted across a range of low and intermediate layers in a pretrained VGG-16 (Simonyan & Zisserman, 2015) and AlexNet (Krizhevsky et al., 2012) CNN could predict human perception of the similarity of images. Building upon this idea Mihai & Hare (2021b) demonstrated that such a loss function could be used to induce a drawing agent to produce sketches that were significantly more interpretable by humans (in the sense of improved agent-human gameplay) than agents trained without such a loss. In the latter case, the agents could learn to play the game well and generalise to unseen images, but the drawings produced by the sender were essentially visual representations of hash codes with rather random sets of lines.

With our modification to move from the VGG16 feature extraction network to the Vision Transformer (ViT) model of the CLIP framework (Radford et al., 2021), we first explored whether a similar

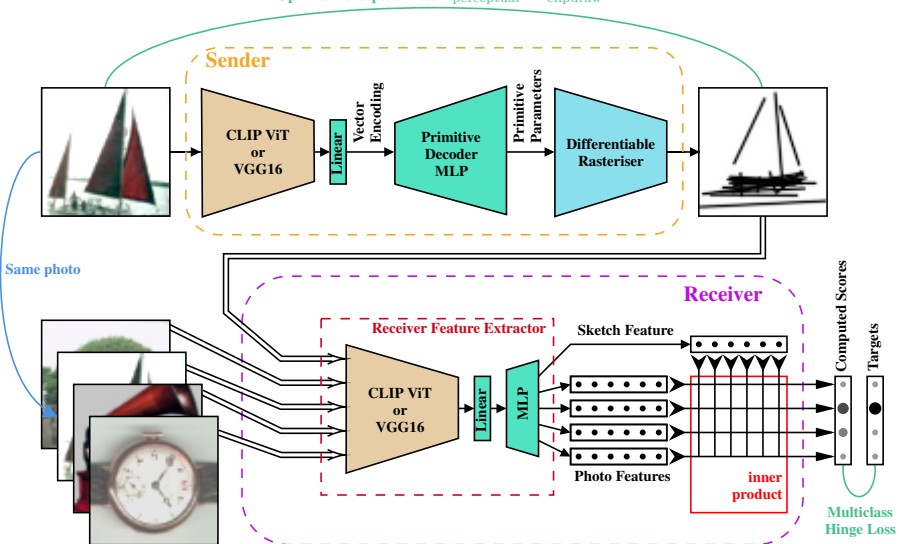

Figure 1: **Model overview.** Two agents are trained to play an image guessing game in which they communicate through a simple line drawing. As indicated, unlike in Mihai & Hare (2021b)'s original model, we also experiment with a more powerful pretrained Vision Transformer encoder module (ViT-B/32 from CLIP (Radford et al., 2021)). An additional perceptual loss (not shown) between the sender's input photo and output sketch induces the sketch to be more understandable. Figure adapted from Mihai & Hare (2021b).

type of perceptual loss to that used by Mihai & Hare (2021b) would be possible. We extracted the feature after each transformer residual block from both the sketch produced by the sender and the corresponding photo that was presented to the sender and used this to compute the loss. The loss itself involves normalising each layer's features, computing the sum squared difference between sketch ($S$) and image ($I$) features at each layer $l$ and performing a weighted sum over the layers, $L$,

$$\mathrm{l}_{\mathrm{perceptual}}(\boldsymbol{S}, \boldsymbol{I}, \boldsymbol{w}) = \sum_{l \in L} \frac{\boldsymbol{w}_l}{n_l} \big\| \hat{\boldsymbol{S}}^{(l)} - \hat{\boldsymbol{I}}^{(l)} \big\|_2^2 \, , \tag{1}$$

where $n_l$ is the dimensionality of the $l$-th layer feature. For the experiments presented here, we used fixed uniform weights for each layer, $w_l = 1 \, \forall l \in L$. The effect of different weights is explored by Mihai & Hare (2021b).

We also investigate another method for generating interpretable drawings, inspired by the approach taken by Frans et al. (2021) The key idea is that we add an extra loss, $\mathrm{l}_{\mathrm{clipdraw}}$, computed by the cosine distance (negative cosine similarity) between the encoded representation, $f(\cdot)$ (e.g. from the last layer of the ViT encoder, or `relu5_3` of the VGG16) of the generated sketch and input image. However, such a loss alone does not result in sketches that are perceptually similar to the input, so instead perceptual similarity is induced by computing the loss over a set of randomly *transformed* sketches, $T$:

$$\mathrm{l}_{\mathrm{clipdraw}}(\boldsymbol{S}, \boldsymbol{I}) = - \sum_{t \in T} \frac{f(t(\boldsymbol{S})) \cdot f(\boldsymbol{I})}{\|f(t(\boldsymbol{S}))\| \|f(\boldsymbol{I})\|} \, . \tag{2}$$

Following Frans et al. (2021), $T$ consists of four randomly sampled transformations created by applying a random perspective transformation and random resizing and cropping in sequence. This crude modelling of physical spatial constraints is sufficient to induce the sketches to be interpretable.

## 3.2 Results

We next present results of the visual communication game played with STL-10 images (Coates et al., 2011) in the original game setup as described by Mihai & Hare (2021b). Figure 2 shows test communication success rates and sketches produced by models constructed with either a VGG16

Table 1: **Human Evaluation results - extended** Trained agents communicate successfully between themselves in all settings. The addition of either perceptual loss allows humans to achieve significantly better than random performance (images from STL-10, original games have 9 distractors/game for these experiments & random chance is 10%). In addition, humans are better at guessing the correct image class when the models are trained with either of the perceptual losses.

| Model | Loss | Agent comm. rate | Human comm. rate | Human class comm. rate |
|---|---|---|---|---|
| VGG16 | $l_{game}$ | 100% | 8.3%($\pm$5.4) | 15.0%($\pm$2.5) |
| VGG16 | $l_{game} + l_{perceptual}$ | 93.3% | 38.3%($\pm$2.5) | 55.6%($\pm$7.1) |
| VGG16 | $l_{game} + l_{clipdraw}$ | 86.7% | 34%($\pm$3.9) | 49.3%($\pm$7.7) |
| ViT-B/32 | $l_{game}$ | 93.3% | 5.6%($\pm$3.1) | 15.6%($\pm$3.1) |
| ViT-B/32 | $l_{game} + l_{perceptual}$ | 96.7% | 45.3%($\pm$5.4) | 63.3%($\pm$7.0) |
| ViT-B/32 | $l_{game} + l_{clipdraw}$ | 96.7% | 62.7%($\pm$11.6) | 83.3%($\pm$9.2) |

or ViT image encoder, trained with only the game objective $l_{game}$, or with the addition of either of the two perceptual losses, $l_{perceptual}$ and $l_{clipdraw}$, described in Section 3.1. For the experiments run with the ImageNet-pretrained VGG16 image encoder, we use the same parameters as specified in Mihai & Hare (2021b). When replacing the image encoder with CLIP's pretrained ViT-B/32 model (Radford et al., 2021), we found that increasing the hidden sizes of the Primitive Decoder, shown in Figure 1, from 64 and 256 to 1024 each, significantly improves the quality of the sketches. It is also worth noting that a bigger learning rate is needed for this model to converge, more specifically 0.001.

Results show that models trained with only the game objective achieve the highest communication success rate, i.e. agents can successfully communicate about the target image, although, they do so by drawing what it looks to us like random sets of lines. The addition of perceptual losses to $l_{game}$ leads to significantly more interpretable sketches.

To assess the level of interpretability, we extended the human evaluation performed by Mihai & Hare (2021b) to include the models studied in this work. This pilot study consists in pairing a pre-trained Sender agent with a human Receiver to play the visual communication game, through a user interface that presents the human with a sketch and 10 possible photographs to choose from. Each human participant played 30 games (i.e. identified 30 sketches) with $K = 9$ distractors for each model configuration. The games are sampled randomly from all those possible within the STL-10 test dataset.

In Figure 2, we include human communication success rates averaged over the 6 participants taking part in this pilot study, for the games played with sketches generated by the corresponding models' Sender agent. Table 1 shows the communication success between the agents playing the games included in this pilot study, the human success rate and an additional measure, the human class communication rate, that looks at the accuracy of humans at determining the class of the sketch rather than the specific instance.

The model using CLIP's image encoder, pretrained on the task of matching (image, text) pairs, leads to better image representations, and eventually, sketches that can be more easily interpreted by humans than those produced by a model with a VGG16 encoder pretrained for the supervised image classification task on ImageNet. Similarly, we observed that the addition of either of the perceptual losses significantly improves humans' ability to recognise the main category depicted in the sketch. More sketches generated during testing can be found in Appendix B.

## 4 Exploring what the drawings mean with prompt engineering

It would be beneficial to be able to understand what information the sender agent is trying to convey through its sketch and compare that to what a human playing the game might try to impart. In the particular game setting we are using, if one communicates only the object in the scene then the expected communication rate would only be 10%. To achieve higher rates much more nuanced information about the image contents needs to be conveyed. Ultimately understanding what is being communicated, and how it differs to humans would allow us to design better approaches to inducing more human-like behaviour in the model and the agent's internal representations.

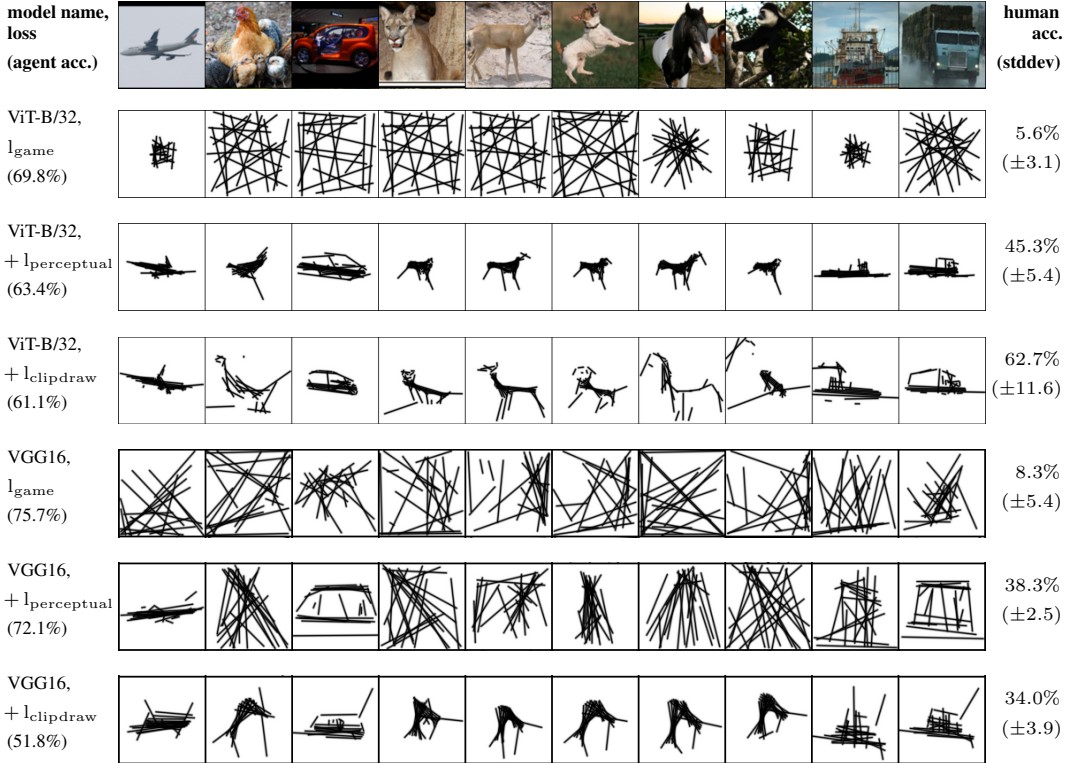

Figure 2: **Sketches from the visual communication game using STL-10 dataset with different image encoders and "perceptual" losses**. Models trained with the $l_{game}$ only do not learn to draw in an interpretable fashion. For both ViT-B/32 and VGG16 image encoders, the addition of either perceptual loss induces more structure into the resulting drawings, making them more similar to the subject of the image, although it decreases that agents' communication success (shown in brackets on the left side). Perceptual losses also quantitatively improve human performance when pitched against agents (note that reported human accuracies on the right-hand side are for games with 9 distractors as opposed to the agent accuracies on the left with 99 distractors). CLIP-pretrained ViT-B/32 models have higher human performance than the VGG models.

To start to explore this in more detail, we demonstrate that we can begin to answer the question of what is being communicated by using the CLIP model as a probe. With the technique of prompt engineering, where a set of textual prompts are encoded with CLIP's language model, it becomes possible to ask basic questions about how CLIP perceives a sketch in terms of the semantic content. For the initial experiments presented here we use two prompt templates: ``a drawing of a XXX.'' and ``a photo of a XXX.''. The placeholder (XXX) is replaced by the 10 different classes in the STL-10 dataset to create a complete set of 20 prompts. For each of the models we then compute a number of statistics regarding CLIPs perception of the sketch, the target image (i.e. the receiver image that is the true answer) and the guessed image (the image that the receiver actually picked), averaged over all 8000 possible games in the STL-10 test set. More specifically we ask which class CLIP perceives an image $I$ to be, using a function $c(I)$ which returns the placeholder of the closest prompt (using cosine similarity in the embedding space), and compare CLIPs predicted class between the sketch, guess and target. In a similar way, we also compute which of the two templates {photo, drawing} CLIP predicts the sketch, target and guess to belong to. Finally, we also utilise a function $gt(input)$ which returns the STL-10 ground-truth class label of the sender agent's input, to allow us to analyse to what extent CLIP's perception of the images matches the true label. The results of this analysis are shown in Table 2.

The results in Table 2 indicate that both forms of perceptual loss do a good job of making the sender agent produce sketches that capture the main class of object in the input image. There is an inherent bias towards CLIP generated sketches because the same model is being used to perform the generation

Table 2: **Comparing models with CLIP using prompt engineering: $c(I)$ returns which class CLIP perceives image $I$ to be; $gt(input)$ returns the true class of the sender agent's input photo; $tp(I)$ returns the type (photo or drawing) CLIP predicts $I$ to be.** There are significant differences between models, however, it is clear that the perceptual losses strongly encourage a more object-centric representation. CLIP is very good at telling the difference between sketches and photos in all cases, despite the perceptual losses pulling together the representations.

| | VGG16 encoder | | | ViT-B/32 encoder | | |
| --- | --- | --- | --- | --- | --- | --- |
| | $l_{game}$ | $+l_{perceptual}$ | $+l_{clipdraw}$ | $l_{game}$ | $+l_{perceptual}$ | $+l_{clipdraw}$ |
| c(sketch)==gt(input) | 7.3% | 24.0% | 41.2% | 9.4% | 96.4% | 96.6% |
| c(sketch)==c(target) | 7.3% | 24.2% | 41.5% | 9.7% | 96.4% | 96.5% |
| c(sketch)==c(guess) | 7.6% | 24.6% | 38.8% | 10.4% | 94.9% | 96.1% |
| c(target)==gt(input) | 97.3% | 97.3% | 97.3% | 97.3% | 97.3% | 97.3% |
| c(guess)==gt(input) | 85.6% | 82.1% | 76.5% | 77.6% | 94.8% | 95.8% |
| tp(sketch)=='drawing' | 100% | 99.9% | 99.9% | 99.9% | 96.2% | 99.3% |
| tp(target)=='photo' | 99.4% | 99.4% | 99.4% | 99.4% | 99.4% | 99.4% |
| tp(guess)=='photo' | 99.4% | 98.4% | 98.4% | 99.0% | 99.4% | 99.4% |

and the probing. The fact that for all models `c(guess)==gt(input)` rates are so high suggests CLIP identifies the class of the guessed image to be correct, i.e. be the same as the ground truth label. On the other hand, `c(sketch)==c(target)` measures if the class CLIP thinks the sketch to be is the same as the class that CLIP thinks the target image to be. The fact that this measure is much lower across all VGG16 models suggests that sketches produced with this feature encoder are not as interpretable to the CLIP model as the sketches produced with the ViT-B/32 encoder. As can be seen in Figure 2, the CLIP generated sketches are qualitatively more interpretable than the VGG ones. When looking at these results bear in mind that the communication game itself is entirely self-supervised; the notion of object class is clearly not required for successful communication and is instead a side effect of inducing a perceptual loss between internal representations. The results also show that despite the perceptual losses forcing the representations of the sketch and image together, the CLIP-based probe is able to recognise the sketch as being a drawing and the receiver images from the dataset as being photos almost all of the time.

## 5   Discussion and Future Directions

The aim of this study was to explore how representational losses influence the sketches produced by artificial agents playing a visual communication game. We showed that the addition of either perceptual loss to the communication game leads to qualitatively more recognisable sketches than those produced by agents trained with the game objective only. Although the additional representational losses slightly decrease the agents' ability to communicate, they significantly increase the possibility to recognise the sketches as the semantic category of the photographs they represent (as shown in Table 2 and with the human evaluation presented in Figure 2). The striking differences between images from the two loss formulations raise lots of questions and this is definitely an area we would like to explore in future work. Clearly, there are many other formulations that would be exciting to experiment with too. Going forwards it would also be interesting to explore if it is possible to minimise the drop in communication rates that arise from introducing this perceptual bias.

Our brief experiments with prompt engineering in the previous section open up a number of doors for future analysis. The game being played by the agents is complex, and the communication success rates are far in excess of the 10% that would naïvely result from the models only communicating information about the class. The obvious next step is to question what additional information is being conveyed in the sketches; is it interpretable semantic information about the input image, or is it some kind of neural hash code that just happens to allow communication to succeed, or is it a mixture of both aspects? Further, similar to the causal interventions that are applied in emergent communication scenarios with non-visual channels, we would also wish to explore which parts of a sketch (perhaps bundles of strokes) contribute to particular aspects of semantic meaning. We hope that with richer datasets and considerably more engineering of prompts that the setup we've outlined in this paper would allow these goals to be achieved.

## Acknowledgments and Disclosure of Funding

D.M. is supported by the EPSRC Doctoral Training Partnership (EP/R513325/1). J.H. received funding from the EPSRC Centre for Spatial Computational Learning (EP/S030069/1). The authors acknowledge the use of the IRIDIS High-Performance Computing Facility, the ECS Alpha Cluster, and associated support services at the University of Southampton in the completion of this work.

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

# Appendices

# A    Model Setup and Training details

For experiments with the VGG16 feature extractor, our setup exactly mirrors Mihai & Hare (2021b):

- STL-10 images are presented to the model at 96x96 resolution
- The final layer VGG-16 features are flattened and projected into a 64-dimension latent space in both sender and receiver agents.
- The sender agent uses an MLP with hidden sizes of 64 and 256 and output layer size of 80 (corresponding to the coordinates end-points of the 20 lines).
- The receiver agent uses an MLP with a 64 dimensional hidden layer and 64 dimensional output.
- all hidden layers use ReLU activations.
- the weights of the VGG16 feature extractor are fixed, and shared between the Sender and Receiver agents.
- Training is done with Adam using and initial learning rate of 1e-4.

For our CLIP-ViT-B/32 modification, VGG16 is replaced with the ViT and everything else remains the same except:

- STL-10 images are rescaled and presented to the model at 224x224 resolution to match the size the ViT was trained on. Both qualitatively and quantitatively making this change for the VGG16 model did not have much effect beyond slowing down training and inference.
- The sender agent uses an MLP with hidden sizes of 1024 and 1024 and output layer size of 80 (corresponding to the coordinates end-points of the 20 lines). We found the model would not work well with the sizes used for the VGG16; conversely training the VGG16 with this increased size didn't qualitatively change the sketches very much, nor affect the communication rate.
- the weights of the ViT-B/32 feature extractor are fixed, and shared between the Sender and Receiver agents.
- Training is done with Adam using and initial learning rate of 1e-3. Using the higher rate seems to help speed up convergence with the ViT. Conversely, with the VGG16 the higher initial learning rate leads to a model that doesn't converge.

# B    Additional Sketching Examples

In Figures 3 and 4 we provide additional examples to better illustrate the difference between the different encoder networks and the effect of including additional perceptual losses on the resulting sketches.

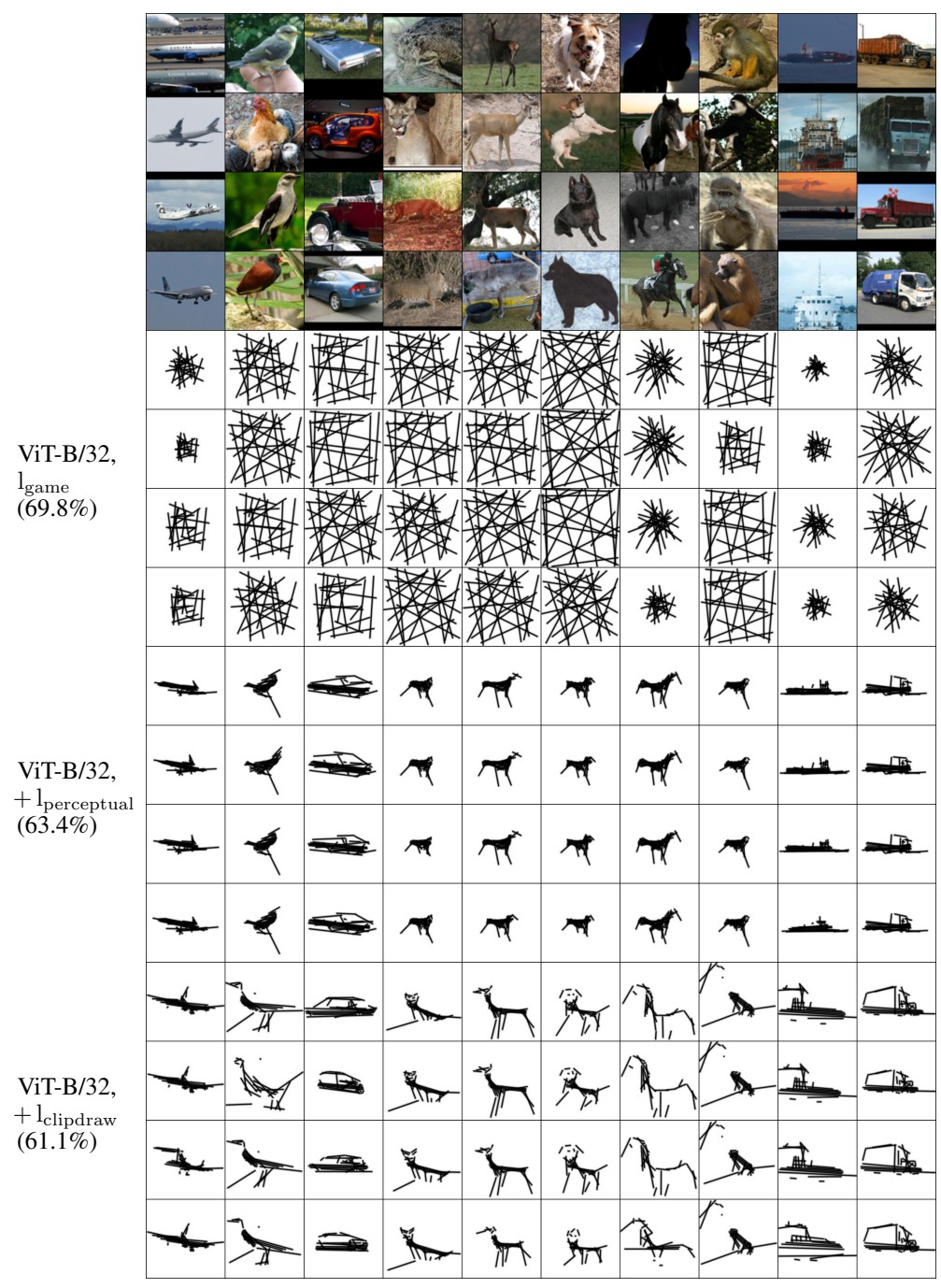

Figure 3: **More sketches of STL-10 test images produced by the model with ViT-B/32 encoder.**

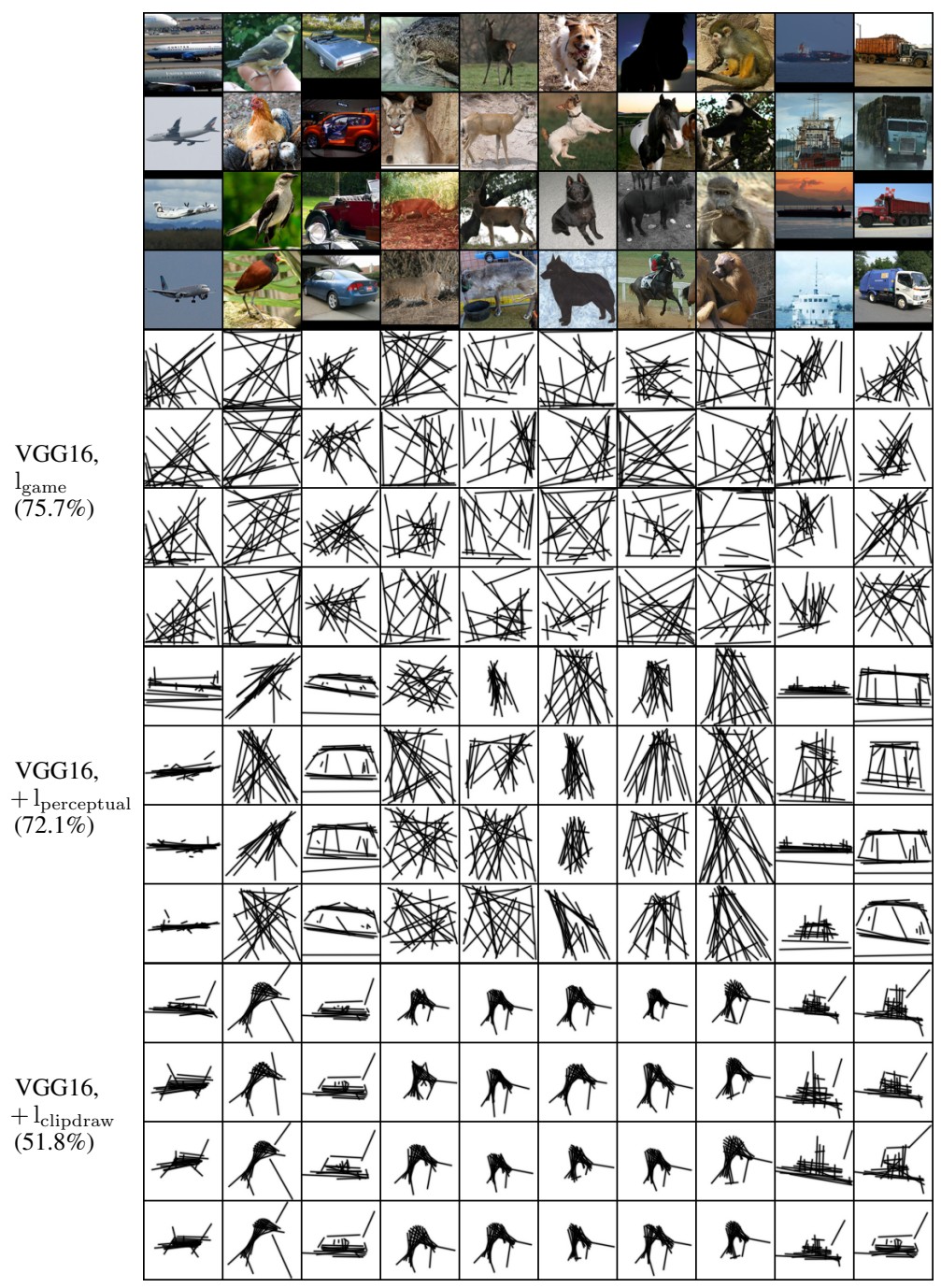

Figure 4: **More sketches of STL-10 test images produced by the model with VGG16 encoder.**

