# OpenReview forum: "Shared Visual Representations of Drawing for Communication: How do different biases affect human interpretability and intent?"
_NeurIPS.cc/2021/Workshop/SVRHM — SVRHM 2021 Poster_

### Official Review · Reviewer_oyov · 2021-10-20
**Review of Shared Visual Representations of Drawing for Communication: How do different biases affect human interpretability and intent? For SVRHM 2021**

**Rating:** 7
**Confidence:** 4

**Review:**

Review of Shared Visual Representations of Drawing for Communication: How do different biases affect human interpretability and intent? For SVRHM 2021

In this paper, the authors explore different improvements to the interpretability of drawings produced by agents in a game of pictorial communication. In this game, one agent had to draw an image and share it with a second agent that needed to guess what it was from a limited pool (that also contained the first agent’s source image). Performance with artificial agents can be very good, but the images produced by the drawing agent appear random to humans. In this paper, the authors attempt to make those images more interpretable by using different loss functions designed to include inductive biases that are receptive to humans. They also compare two different network architectures.

I found the paper well written, interesting, and appropriate in scope for SVRHM. I recommend it for the workshop.

Some suggestions to consider:

- My main complaint is that there are many claims throughout the paper about interpretability with no human data to support those claims. Wherever the authors have said that the drawings are more interpretable, we are relying on their intuition. E.g. pg 3 line 105. For what it’s worth, I mostly agree with their characterization, but it is an anecdote/speculation. Consider that ViT-B with the new loss functions are both described as more interpretable -- what if we were to find that these images do not actually help a human identify the sender’s source image? Or that they do so with different levels of efficacy? Would we still say that they are both interpretable? The claims in this paper could be significantly strengthened by validating the method with human data. A simple online task where naive raters attempt to pair the source images with the agent’s drawings in different conditions. This human performance could then be compared to the agent performance, or against chance (performance expected by random guessing). The results of this validation task could be displayed with Figure 2.
- I think VGG16 with l(perceptual) is not very interpretable, and I bet if we conduct the human validation study described above, it would not match human performance. Moreover, VGG16 + l(clipdraw) seems to converge on only two different sketch forms (Figure 4).
- I think it might be helpful to adapt FIgure 1 to illustrate where the new inductive biases occur
- I had a hard time understanding the message of section 4 and the associated results in Table 1. I understand the notion of prompt engineering and the goal of the exercise, but expressing the results in terms of c(I)==etc… without a header or guide to interpreting those results made it difficult.

---

> ### Author Response · Authors · 2021-12-04
> **Response to Reviewer oyov**
>
> We would like to thank the reviewer for their time and for the suggestions. We respond to specific comments below:
>
> >My main complaint is that there are many claims throughout the paper about interpretability with no human data to support those claims.
>
> We again thank the reviewer for pointing this out, we respond to this suggestion in the comment addressed to all reviewers.
>
> >I think VGG16 with l(perceptual) is not very interpretable, and I bet if we conduct the human validation study described above, it would not match human performance. Moreover, VGG16 + l(clipdraw) seems to converge on only two different sketch forms (Figure 4).
>
> As indicated by the human evaluation study, the VGG16 with the perceptual loss has a $38.3 (\pm2.5)$ human success rate and it generates statistically more interpretable sketches than the same model trained with only the game loss. As for the VGG16 trained with the additional clipdraw loss, we have now reorganised the figures to include examples from all possible classes. Figure 4 now shows that although the model does not have a very distinctive approach for representing animals, the remaining 4 classes have visually different sketched representations.
>
> >It might be helpful to adapt Figure 1 to illustrate where the new inductive biases occur
>
> This is a good suggestion! We have now added it to the model diagram.
>
> > Expressing the results in terms of c(I)==etc… without a header or guide to interpreting those results made it difficult
>
> This is a fair point. We have now added a guide on how to interpret the results in the table caption and extended the discussion.
>
> We once again thank the reviewer for their time and specific suggestions, it definitely helped us strengthen the paper. We hope we addressed all the points and answered all the questions raised by the reviewer.

---

### Official Review · Reviewer_8222 · 2021-10-27
**Ideas of interest, some confusing details**

**Rating:** 7
**Confidence:** 4

**Review:**

This paper investigates how different loss functions affect the output of networks playing a sketch communication game. The ideas are of interest to the SVRHM community and I believe the paper should be accepted.

My primary comment is that the paper would be significantly strengthened by including human evaluations of the understandability of sketches. The paper makes various assertions about how understandable the sketches are for humans, and while the example sketches are certainly interesting, I would not be surprised if human accuracy in labelling the sketches across a wide set of stimuli is actually not particularly high. Some of the sketches in Figures 2, 3 and 4 are introspectively difficult to interpret without also having seen the target image.

A similar assertion that could be tempered is on lines 58--60. We don't know how hard this game would be for humans. For example, humans can remember specific object images with surprisingly high accuracy (https://www.pnas.org/content/105/38/14325).

## Minor comments

- Introduction would benefit from an additional opening sentence setting the problem into context. Why study this?
- line 79: it's a bit confusing to formulate the loss with a weight that is then not used (all set to one).
- line 95: the game objective is not explicitly defined. Is it accuracy? likelihood?
- Figure 2: The examples show that it's not simply a perceptual vs game loss: VGG16 with the perceptual loss also fails to produce human-interpretable sketches. This should be more prominently discussed.
- The expected communication rate (line 114) is not really meaningful for sketches, in my opinion. The baseline in a verbal game would be 10%, since there it's possible to only communicate the object label (e.g. "dog") but in human sketches, I find it difficult to imagine communicating the concept of "dog" without also including additional information (e.g. pose and viewpoint), which would increase discriminability of the candidate image set (i.e. raise the baseline).
- Table 1: This table is confusing to interpret and could be better explained in text (lines 118--132). For example, what does it mean that the c(guess)==gt(input) rates are so high (particularly compared to the c(sketch)==c(target) for VGG with perceptual losses)? Does this mean that labels based on sketches are significantly worse than guessing in this setting? If so, this needs further explanation.

---

> ### Author Response · Authors · 2021-12-04
> **Response to Reviewer 8222**
>
> We would like to begin by thanking the reviewer for their time and for the suggestions to help strengthen our work! We respond to specific comments below:
>
> > The paper would be significantly strengthened by including human evaluations
>
> We thank the reviewer for this suggestion, we address this comment in the response to all reviewers.
>
> >Lines 58--60. We don't know how hard this game would be for humans
>
> As also answered to Reviewer CsmL, we tested this game with human participants and observe what we speculated, this game setting in which humans have to pick a target from a list of 100 possible images (99 are distractors), from which multiple images could be from the same class as the target is very difficult to solve with a constrained communication channel. We can imagine that if additional characteristics of the target image would be included in the drawing, such as colour, it might help the human differentiate between multiple instances, but the scenario presented in this work is not feasible. Hence when we perform the human evaluation, we do so using 9 distractors only.
>
> > Introduction would benefit from an additional opening sentence setting the problem into context
>
> We agree, we have now added to the opening of that section an introduction to the problem context. Thanks for pointing that out!
>
> > Line 79: it's a bit confusing to formulate the loss with a weight that is then not used (all set to one)
>
> This is a fair point, we do not explore the effect of weighting $l_{perceptual}$ in this work, but because it is the same as the one used in our other work [A1], we didn't want to confuse by changing its formulation. We have now added a sentence that explains that this factor is explored in the original paper.
>
> >line 95: the game objective is not explicitly defined
>
> From the model diagram, it is the hinge loss between computed scores (inner product between sketch features and the features of all possible photographs) and targets. We have now added this explicitly in the text.
>
> >Figure 2: The examples show that it's not simply a perceptual vs game loss: VGG16 with the perceptual loss also fails to produce human-interpretable sketches.
>
> We agree that the sample sketches for VGG16 with the perceptual loss might not seem relevant, but they have to be taken in the context. As observed in [A1] the model, with this number of line strokes, is not representing classes of animals very well, but it starts capturing the structure. On the other hand, the model does well on classes like boats/cars/trucks/airplanes. The result must be seen in the context of the other paper, which shows that compared to a model trained with only the game loss, the addition of a perceptual loss significantly increases humans' ability to guess the correct target image as demonstrated in the human evaluation.
>
> > The baseline in a verbal game would be 10%, but in human sketches, I find it difficult to imagine communicating the concept of "dog" without also including additional information...(i.e. raise the baseline).
>
> Yes, the assertion about baseline in a verbal game with the vocabulary restricted to the class names and 99 distractors would be 10%. Clearly, additional information is being encoded in the sketches and allows the agents (and to some extent humans) to predict the correct image. Pose and viewpoint would likely be factors that humans would use to communicate extra information, although as noted in the question/response to reviewer CsmL orientation does not seem to be utilised by the agents, perhaps as a result of a dataset bias, or as a result of them having learned a "better" alternative.  This "better" alternative might however not be very human (indeed that seems likely the case looking at the sketches produced). Encouraging an even more human-like way of communicating very much remains an open research question, and requires a lot more understanding of how these current models are actually working.
>
>
> > For example, what does it mean that the c(guess)==gt(input) rates are so high (particularly compared to the c(sketch)==c(target) for VGG with perceptual losses)? Does this mean that labels based on sketches are significantly worse than guessing in this setting? If so, this needs further explanation.
>
> Sure, we hope we've now explained this better in the text. The fact that c(guess)==gt(input) rates are so high suggest CLIP identifies the class of the guessed image to be correct, i.e. be the same as the ground truth label. On the other hand, c(sketch)==c(target) measures if the class CLIP thinks the sketch to be is the same as the class that CLIP thinks the target image to be. The fact that it is much lower suggests that sketches produced with a VGG16 feature encoder are not very interpretable to the CLIP model, which supports what one can also observe by looking at the sketches.
>
> [A1]: Mihai, Daniela and Hare, Jonathon (2021). Learning to draw: Emergent communication through sketching. NeurIPS 2021. URL https://bit.ly/3G28PSn

---

### Official Review · Reviewer_Dwvm · 2021-10-28
**Good paper, some edits needed**

**Rating:** 7
**Confidence:** 4

**Review:**

Summary: \
While prior work has shown that it is possible to train a pair of neural networks to come to common ground in a sketching communication game, the sketches the sketcher network makes are meaningful to the receiver network but not to humans. This paper explores how representational biases imposed on such networks can make the sketches more ‘human-like’.
The first contribution is extending prior work by Mihai & Hare by using the recently released CLIP architecture. The sketcher network is forced to make its representations more image like due to the imposition of a ‘perceptual loss’ between the network’s feature representations for the sketch and the target image. They also compute a loss between features from the final layer of the sketcher networks for the target image and transformed versions of the sketch created by the sender. This too increased perceptual similarity between the sketch and target.

The authors also offer an interesting prompt engineering based approach to explore if the models can match text prompts to the actual image shown.
It isn’t surprising that the models can discriminate between the sketch and image modalities due to their very different pixel distributions. However, this result continues to show that the ‘clipdraw’ loss is important for aligning text-based descriptions.

Comments: \
This paper raises some very interesting points and raises important questions about the tradeoffs between perceptual fidelity and communicative efficiency in communication games. However, many of the experiments and architectural elements of the models in this paper refer to the model presented by Mihai & Hare in prior work. Even Figure 1. appears to be a slight modification of a figure from said paper. Without any information about how the models ‘play’ the communication game and adequate motivation as to why this is a sensible way to model visual communication it is difficult to evaluate parts of the paper. I am left with questions such as – “Does the sketcher and receiver model have the same encoder weights?”, “Is the receiver’s guess chosen based solely on the softmaxed dot-product between sketch and image features?”, “How many items of each class are present on each trial? (I think the text implies 10 of each, but it is not explicitly stated)”.
I would recommend shortening the introduction to add some details about Mihai & Hare’s model or to flesh out the Model Setup and Training Details section in the appendix.
Additionally, some commentary on why the clipdraw loss seems to be better than the perceptual loss would be nice to include.

Finally, it would be interesting to follow up with some human experiments. There are many papers that might prove to be helpful references for doing so. Here are some to consider for future work – \
RD Hawkins, M Sano, ND Goodman, J Fan (under review). Visual resemblance and communicative context constrain the emergence of graphical conventions. - https://cogtoolslab.github.io/pdf/hawkins_arxiv_2021.pdf

RD Hawkins, M Kwon, D Sadigh, ND Goodman (2020). Continual adaptation for efficient machine communication. - https://www.aclweb.org/anthology/2020.conll-1.33/

JE Fan, RD Hawkins, M Wu, ND Goodman (2020). Pragmatic inference and visual abstraction enable contextual flexibility during visual communication. - https://rdcu.be/bQqxr


Rationale for Score: \
The importance of making the outputs of artificial communicative agents more human-like is a promising research program and the results would be of general interest to the community. The paper could do with a few additional details about the base model and the motivations behind the perceptual loss functions, but these can be addressed fairly easily in the camera-ready version. Overall a nice paper.

---

> ### Author Response · Authors · 2021-12-04
> **Response to Reviewer Dwvm**
>
> Firstly, we would like to thank the reviewer for the extensive feedback and for the suggestions to strengthen our work. We respond to specific comments below:
>
> > Many of the experiments and architectural elements of the models in this paper refer to the model presented by Mihai & Hare in prior work [...]. Without any information about how the models ‘play’ the communication game [...] it is difficult to evaluate parts of the paper.
>
> We agree with this comment, thank you for the questions. This work is based on our previous work, and because of space constraints, we have left out details about the framework which we considered would be explained in the referenced work. However, we now answered your questions by extending section 3 and adding to the Training Details Appendix.
>
> >> Does the sketcher and receiver model have the same encoder weights?
>
> Yes, for the purpose of this study the encoder used is either pretrained on ImageNet in the case of VGG-16 or on CLIP's (text, image) pairs in the case of ViT-B/32 image encoder and the weights are frozen during gameplay.
>
> >> Is the receiver’s guess chosen based solely on the softmaxed dot-product between sketch and image features?
>
> No, not quite; there is no softmax involved. Following earlier models [A1] the largest dot product indicates the guess, and a multi-class hinge loss is used during training (although quick experiments showed that the model still works well with a more common NNL loss over the softmax probabilities). Because the target vector is normalised this is essentially equivalent to picking the image with the closest cosine similarity. Note that we could induce a distribution over the outputs with a softmax, but that it wouldn't change the guessed image.
>
> >> How many items of each class are present on each trial? (I think the text implies 10 of each, but it is not explicitly stated)
>
> For the game setup used in this paper, the items presented to the receiver are sampled randomly from all STL-10 training set, 99 distractors + target. The expected number of items from each class is 10, but we do not enforce it. We now also include results with a human receiver which were played with 9 distractor images.
>
> >Some commentary on why the clipdraw loss seems to be better than the perceptual loss would be nice to include.
>
> Thank you for the suggestion! As mentioned to reviewer CsmL, we aim to explore this topic further in future work and look into other loss perceptual loss formulations.
>
> >Finally, it would be interesting to follow up with some human experiments.
>
> Thank you, we have taken this into consideration and since we have already done this for the previous work, we extended the human evaluation to include models presented in this work too, as mentioned in our response to all reviewers. Results are now included in the paper.
>
> [A1]: Havrylov, Serhii & Titov, Ivan. (2017). Emergence of Language with Multi-agent Games: Learning to Communicate with Sequences of Symbols. NIPS 2017

---

### Official Review · Reviewer_CsmL · 2021-11-01
**Shared Visual Representations of Drawing for Communication: How do different biases affect human interpretability and intent?**

**Rating:** 4
**Confidence:** 3

**Review:**


# Quality & Clarity: 5

This work investigate several inductive biases in the context of communication games involving visual interpretability.

In the introduction and background section, the authors describe previous computational approaches clearly, establishing the lineage of the current work under Mihai & Hare 2021.

However, on line `24` the introduction fails to clearly state the primary contribution of this work, instead opting for deflationary rhetoric of
> In this paper we explore Mihai & Hare (2021b)’s model in more detail.
Without a clear statement of purpose (i.e. "Here, we explore the interaction of more "human-like" perceptual representations in communication with...") the remainder of this paragraph appears to be arbitrary compositions of other works in the field of communications and computer vision.

In the background, starting on line 40, the authors establish a link between human vision's ability to represent scenes at different levels of representation. However, this link was disorganized. Instead of enumerating cited works at the end of the paragraph, the manuscript's clarity would have been dramatically improved if this sections began with human vision as an inspiration and then referenced the relevant communication research.

In the model section, lines 57-60 mention that the communication via drawing task that the model is trained to perform "seems impossible for humans" without any citation or further explanation. Humans already enjoy several board games that rely on drawing real-world objects. Without a stronger qualification than intuition this comment is distracting at best or wrong at worst. It is unclear how framing the relevant task as super-human benefits the work.

# Originality NA

This reviewer is not familiar with the literature on communication and does not have a comment for originality.

# Significance 5

Understanding and implementing the computational principles that allow humans to effectively communicate under both processing and pragmatic constraints is a central aspect to human intelligence.

This work aims at establishing a set of working components: 1) a visual system that can effectively represent the world in a way that 2) summarizes key features that 3) can be queried meaningfully.

From this manuscript, 1 & 3 are given partial service but the degree to which these goals are accomplished are obfuscated by the format of Table 1.

It is unclear that this work achieves goal 2. In Fig 2, the authors illustrate the qualitative interpretability of model sketches from a variety of images. It is  clear that adding additional inductive biases in the form of perceptual losses leads to differences across models where those that include such losses have recognizable drawings (especially row 3: ViT-B/32 + clipdraw). However, as the authors mention in future directions, it is unclear whether the model is drawing the scene or collapsing to the scene category. The authors only show two scenes that are close in category (horse, horse + jockey) and their resulting sketches do not appear to reflect any interpetable difference; both, sketches appear to be a horse. The additional sketches on page 9 do no alleviate this concern, with objects in several sketches oriented in the reverse direction than those in the input image. Given that this work relies on qualitative comparisons to support the conclusion of the title, it would be prudent to organize such figures by category such that this could be easily illustrated to readers.


# Pro / Cons

## Pros

- important premise: Human vision can represent key aspects of a scene succinctly
- exploration of novel model by composing CLIP and prompt engineering

## Cons
- paper itself suffers from lack of clarity and structure
- it is unclear that the authors meet their objective of qualitative interpretability based on the Fig 2
- it is unclear why the different approaches in the work lead to differences in performance in terms of interpretability.

---

> ### Author Response · Authors · 2021-12-04
> **Response to Reviewer CsmL**
>
> We would like to thank the reviewer for their time and feedback. We respond to specific comments below:
>
> > On line 24 the introduction fails to clearly state the primary contribution of this work...
>
> We have now included a clear statement of purpose to begin that paragraph with. To reiterate, this work's contribution is in the use of perceptual losses to constrain a visual communication channel to capture representations of the visual world shared between humans and machines.
>
> > The manuscript's clarity would have been dramatically improved if this sections began with human vision [...].
>
> We agree with this suggestion and have now re-organised the order of ideas in this paragraph. Thanks for that!
>
> > Lines 57-60 mention that the communication via drawing task that the model is trained to perform "seems impossible for humans" without any citation or further explanation.
>
> With that paragraph, we wanted to emphasise the original game setup which is used in this paper, played with $K=99$ distractors sampled from 10 possible classes of STL-10. The list of distractors will include more images from the same class as the target in this setting. Hence, even if a human can identify correctly the class of a sketch, they will still have difficulty in recognising the correct target without additional information. The focus of this work is on a **constrained** communication channel. We do not disagree with the reviewer's claim that people can successfully play board games that rely on drawing real-world objects. However, when asking a human receiver to play the game configuration that has got the highest average communication rate (that is ViT-B/32 encoder with the additional $l_{clipdraw}$) with 99 distractors instead of 9, the communication rate drastically decreases to $6.7\%$. This shows that without additional discriminative details, the exact target cannot be identified amongst that many distractors.
>
> > ... these goals are accomplished are obfuscated by the format of Table 1.
>
> We appreciate that Table 1 might be more difficult to interpret, but we considered it to be the "cleanest" format for summarising the prompt engineering experiment. To help the interpretation of results, we now re-state what each of the queries means in the caption of the table. We also extended the discussion to give more insight into what the results shown in that table mean.
>
> >The authors only show two scenes that are close in category (horse, horse + jockey) and their resulting sketches do not appear to reflect any interpretable difference.
>
> We would like to clarify this confusion, the two images belong to the same category, horse. We updated that figure to include an example from each class to avoid this confusion. As for the model collapsing to a scene category, we conjecture that is a result of the constraints applied on the communication channel (i.e. limited number of strokes, capacity visual encoder/primitive decoder).
>
> >  The additional sketches on page 9 do not alleviate this concern, with objects in several sketches oriented in the reverse direction than those in the input image.
>
> That again is a consequence of the model learning one sort of representation for all possible images in that class. The way that the agents learn to distinguish multiple instances of the same class has to do with more subtle details than orientation. For a human, however, we agree that orienting the sketch in the same direction as the target photograph can help. However, that depends heavily on the context given to the receiver [A1] - for example, if the dataset is biased with respect to direction, there might be better ways of communicating the correct image.
>
> > Given this work relies on qualitative comparisons [...] it would be prudent to organise such figures by category.
>
> Thank you for this suggestion, we now organise the figures by category.
>
> > It is unclear that the authors meet their objective of qualitative interpretability based on Fig 2
>
> As per our answer to all reviewers, we make these assumptions in light of the paper this work is also based on. Based on our observation, we claim that the perceptual loss inspired from [A2] are more interpretable, but we now back this up with human data.
>
> >It is unclear why the different approaches in the work lead to differences in performance in terms of interpretability.
>
> We agree, and this is definitely an area we would like to explore in future work. The striking differences between images from the two loss formulations raise lots of questions. Clearly, there are many other formulations that would be exciting to experiment with too.
>
> [A1] JE Fan, RD Hawkins, M Wu, ND Goodman (2020). Pragmatic inference and visual abstraction enable contextual flexibility during visual communication. - https://rdcu.be/bQqxr
>
> [A2] Kevin Frans, Lisa B. Soros, and Olaf Witkowski. Clipdraw: Exploring text-to-drawing synthesis through
> language-image encoders. CoRR, abs/2106.14843, 2021. URL https://arxiv.org/abs/2106.14843

---

### Author Response · Authors · 2021-12-04
**Response on All Reviewers**

We want to thank the reviewers for their time and their valuable feedback. We appreciate the specific suggestions and thoughtful questions which can definitely strengthen our paper. We address each reviewer’s comments within replies to each review. We will incorporate all feedback in the final updated version of the paper.

## Pilot Study - Human Evaluation

We note that all reviewers have asked for human evaluation data to back up the claims related to the interpretability of the sketches. We would like to clarify that this aspect has been dealt with in the updated version of the paper that this one builds upon. We thus kindly invite the reviewers to have a look at the updated version of [A1]. However, we extended the human evaluation experiment performed for that to include the models presented in this paper, and we now include the results of this pilot study as described below.

We follow the setup described in [A1] and pair a human receiver with a pretrained sender agent. We present the human participant with sketches produced by a sender pretrained under different conditions (with different training objectives and visual encoder networks). Each participant plays 30 games per setting - i.e. tries to identify the correct target from a grid of 10 possible images, based only on the sketch. The games are chosen randomly from all those possible within the STL-10 test dataset.

Participants receive no feedback about their performance. We set the experiment to be played with $K=9$ distractors because it gives the humans a plausible chance to play the game. We address comments regarding the difficulty of playing the game with $K=99$ distractors in individual comments to Reviewer CsmL and Reviewer 8222. This pilot study is performed with 6 participants and we provide standard deviation along with the human communication success rate which measures the success at selecting the correct target image.

Results are added in Figure 2 of the paper and extended in Table 1 for each model configuration. As expected, the ViT-B/32 model with the additional clipdraw loss draws the most interpretable sketches according to the human evaluation, followed by the same model trained with the perceptual loss. In the case of either image encoder network, the models trained with either of the 2 perceptual losses led to a statistically significant improvement in humans' ability to play the game successfully.

[A1]: Mihai, Daniela and Hare, Jonathon (2021). Learning to draw: Emergent communication through sketching. NeurIPS 2021. URL https://bit.ly/3G28PSn

---

### Decision · Program_Chairs · 2021-11-02

Accept (Poster)